# The Protective Effects of Astaxanthin (AST) in the Liver of Weaned Piglets

**DOI:** 10.3390/ani13203268

**Published:** 2023-10-19

**Authors:** Kinga Szczepanik, Maria Oczkowicz, Piotr Dobrowolski, Małgorzata Świątkiewicz

**Affiliations:** 1Department of Animal Nutrition and Feed Science, National Research Institute of Animal Production, Krakowska St. 1, 32-083 Balice, Poland; kinga.szczepanik@iz.edu.pl (K.S.); malgorzata.swiatkiewicz@iz.edu.pl (M.Ś.); 2Department of Animal Molecular Biology, National Research Institute of Animal Production, Krakowska St. 1, 32-083 Balice, Poland; 3Department of Functional Anatomy and Cytobiology, Maria Curie-Skłodowska University, Akademicka St. 19, 20-033 Lublin, Poland; piotr.dobrowolski@umcs.lublin.pl

**Keywords:** astaxanthin, pigs, weaned, liver, oxidative stress, collagen

## Abstract

**Simple Summary:**

Weaning is a period when young animals are exposed to a number of stressors caused by the separation from the sow, transport and handling, different food sources, a new social hierarchy, and different physical environments. This situation contributes to the generation of free radicals that cause oxidative stress. In weaned piglets, the most prominent disorders occur in the intestines, and their increased permeability can compromise liver function as metabolites migrate to the liver. Oxidative stress can lead to liver necrosis. Astaxanthin (AST) is a powerful carotenoid and antioxidant that can scavenge free radicals and protect cells from oxidative damage. Findings show that AST has a protective effect on the liver, reducing inflammation, lipid accumulation and markers of liver damage. It also inhibits insulin resistance and lipotoxicity-induced steatohepatitis and regulates fatty acid metabolism pathways. This study investigated the liver-protective properties of AST in weaned piglets. The study showed that AST reduces collagen content in liver tissue and affects the expression of specific genes related to liver function. Expression of CREB, NOTCH1 and NR1H3 genes decreased, while expression of the CYP7A1 gene increased. These findings underscore the beneficial effects of AST on liver health and suggest its potential as a protective agent against liver damage.

**Abstract:**

During the weaning period, piglets are exposed to high levels of stress, which often causes problems with the digestive system. This stress also promotes the production of free radicals, resulting in oxidative stress. Astaxanthin (AST) stands out as one of the most potent antioxidants. Its resistance to light and heat makes it particularly valuable in compound feed production. This study was to determine the effect of AST impact on liver histology and gene expression in piglets. For our experiment, we used 16 weaned piglets of the PL breed, which we divided into two groups: Group I (control group with no AST supplementation) and Group II (supplemented with AST at 0.025 g/kg). Both feed mixtures were iso-proteins and iso-energetic, meeting the nutritional requirements of the piglets. The experiment lasted from day 35 to day 70 of the piglets’ age, during which they had ad libitum access. The results indicate that the addition of AST prevents liver fibrosis due to reduced collagen deposition in the tissue. Analysis of gene expression supported these results. In the AST-supplemented group, we noted a decrease in *NR1H3* expression, an increase in *CYP7A1* expression, and reductions in the expression of *NOTCH1* and *CREB* genes.

## 1. Introduction

Astaxanthin (3,3′-dihydroxy-β,β-carotene-4,4′-dione) belongs to the xanthophyll carotenoid group [1]. It is found in marine organisms such as shrimps, crabs, fish, and algae, and is also found in yeast and bird feathers [2]. It is characterized by its properties such as antioxidant, anti-inflammatory and anticancer abilities [1,3]. Acting as a potent antioxidant, astaxanthin serves as a free radical scavenger and reactive oxygen species (ROS) quencher, thereby protecting molecules and cellular membranes from oxidative damage [4,5]. Free radicals are formed either by both normal cellular metabolism and external sources like pollutants, radiation, and drugs. When there is an inability to gradually destroy excess free radicals, their accumulation in the body leads to the formation of oxidative stress [6]. Oxidative stress plays a role in the pathogenesis of many diseases associated with liver damage [1]. The liver, being the largest gland, performs detoxification, metabolic, filtration and storage functions. According to studies, oxidative stress mediates the progression of fibrosis, and molecules associated with oxidative stress can also act as mediators for molecular and cellular events associated with liver fibrosis [7]. Stellate cells (HSC), primarily responsible for liver fibrosis, along with Kupffer cells and attracted mononuclear cells, play significant roles in this process [7,8]. The interactions are controlled and induced by chemical mediators, among which transforming growth factor β (TGF-β) plays a special role. TGF-β expression and synthesis are modulated mainly by redox reactions [9]. One of the main pathways underlying fibrogenesis in the liver is the activation of HSC due to organ damage. During activation, HSC transforms from resting to proliferating and contracting myofibroblast-like cells and produces type I collagen, which is one of the main components of the extracellular matrix [10,11,12]. Oxidative radicals, by impairing mitochondrial function, can instigate hepatocyte death through various means, such as oncotic necrosis, necroptosis, and apoptosis, and promote the secretion of proinflammatory mediators [13]. Consequently, ROS are believed to potentially foster fibrosis by damaging hepatocytes or by activating Kupffer cells and HSC [10,14].

Scientific reports suggest a protective effect of AST on the liver. In a study by Sila et al. [15], the potential of astaxanthin to counteract diabetic complications was explored in adult rats. The results underscored its capability to reverse hepatotoxicity. Liu et al. [2] showed an effect of AST on the intestinal microflora of mice. Their study revealed that AST notably attenuated inflammation and reduced excessive lipid accumulation and serum markers of liver damage. Another study showed that this carotenoid inhibited and reversed mouse lipotoxicity-induced insulin resistance and steatohepatitis, attributed to the reduction in hepatic lipid accumulation and oxidation [16]. Zhang et al. [17] further cemented AST’s hepatoprotective effects through their experiments on Wistar rats exposed to arsenic exposure. AST treatment attenuated liver damage induced by long-term arsenic exposure reduced the increase in inflammatory cytokines like NF-kB, tumor necrosis factor-α, and interleukin-1β, as well as lowered oxidative stress levels and overall arsenic content in the liver. In an experiment by Jia et al. [18], AST was shown to have a beneficial effect on hepatic steatosis. The research established that astaxanthin elevates *PPAR* (both *PPARα* and *PPARγ*) levels and impedes the Akt-mTOR pathway, subsequently influencing lipid metabolism in the liver [18].

In young piglets during the weaning period, the most described health problems are due to intestinal dysfunction. They result from impaired nutrient absorption caused by, among other things, atrophy of the intestinal villi and an abnormal ratio of villi length to crypt depth [19]. Little attention during this period is focused on the liver, which is extremely susceptible to oxidative stress. In addition, blood reaches the liver from the intestines through the portal vein; thus, when the intestinal function is disrupted, increased intestinal permeability can contribute to the translocation of metabolites to the liver and impaired liver function [20]. A large amount of oxygen is consumed in the liver, which is related to the presence of numerous mitochondria. This, in turn, is associated with the formation of ROS. Mitochondria metabolize about 80–90% of the oxygen used by hepatocytes and 2% of the oxygen consumed is converted to superoxide anions [21,22]. However, despite the liver’s susceptibility to various stress forms, there is a noticeable research gap regarding liver dysfunction resulting from oxidative stress in weaned piglets [23].

Our study was designed to assess the impact of AST on the liver health of weaned piglets, both histologically and genomically. Building on the findings of Szczepanik et al. [24] that confirmed the beneficial effects of AST on liver panel biochemistry and blood morphology in weaned pigs, the present study was extended to analyze the protective effect of AST supplementation on piglet livers. Our hypothesis postulated that AST, given its potent antioxidant properties, would reduce oxidative stress, consequently shielding the liver from excessive collagen deposition amidst liver lobules. This, in turn, would prevent possible pathological fibrosis. For gene expression analyses, we selected genes for encoding receptor proteins (*NOTCH1*), genes that are regulators of key liver functions, such as fatty acid metabolism (*ACAA2*, *CYP7A1*), lipids (*NR1H3*), sugars (*HK1*), cholesterol metabolism (*SREBF2*, *APOE*, *DHCR24*), mitochondrial mechanisms (*SIRT3*), and exogenous chemical metabolism (*CYP1A1*). Additionally, our analysis encompassed genes either directly or indirectly related to extracellular matrix deposition (*CREB1*, *COL1A2*), immune reactions (*CXCL10*) and antioxidant protection (*SOD1*, *CAT*). Histopathological analysis, in turn, allowed us to assess the percentage of collagen in the tissue and predict the antifibrotic effect of AST.

## 2. Materials and Methods

### 2.1. Ethical Approval

All experimental procedures involving live animals in this study were conducted in strict accordance with the guidelines set forth by the first Local Ethics Committee for Experiments with Animals in Cracow, Poland (Resolution No. 420/2020, dated 22 July 2020). During the experimental period, the health of the postweaning pigs was regularly monitored by a veterinarian.

### 2.2. Animals

The experiment was conducted on sixteen 35-day-old postweaning barrows of the Polish Landrace (PL) breed, each weighing approximately 8.7 kg (±0.2 kg). These pigs were divided into two distinct groups, with each group containing eight pigs. Group I served as the control, while Group II received astaxanthin supplementation. The source of astaxanthin was *Haematoccocus pluvialis* (Podkowa AD 1905 sp. z o.o., Lublin, Poland), and was added in the amount of 0.025 g/1 kg (25 mg/kg) of feed mixture. It is noteworthy that AST has not yet been recognized in pig nutrition, resulting in a limited number of studies in this area. The dose of AST for the piglet feed in our experiment was determined by referencing a few existing piglet studies [25]. Furthermore, we considered the observed higher antioxidant activity of AST when compared to vitamin E [26,27], an antioxidant routinely incorporated into pig feeds. All the pigs were given a diet that was both iso-protein and iso-energetic, adhering to the nutritional standards set by the Polish standards of pig feeding [28]. Table 1 presents a detailed breakdown of the diets’ ingredient composition and basic chemical analyses of feed mixture samples were performed according to standard methods [29].

The experimental fattening phase lasted 35 days. During this period, the pigs were housed individually in pens, each fitted with a self-feeder and nipple waterers providing ad libitum access to feed and water. The pigs were kept in a mechanically ventilated room, maintaining a temperature of 18–20 °C. However, during the initial acclimatization phase, heating lamps were utilized to elevate the surrounding temperature to approximately 28 °C. The room’s air humidity averaged around 55%, with an air exchange of 15×/h, and a natural day/night light cycle was maintained. The pigs were kept in individual pens and received feed and water ad libitum. At the end of the experiment, all pigs were slaughtered. The animals were killed with an approved standard method by simply stunning them with a specialized penetrating pin device Blitz (Bad Neustadt, Germany) designed for slaughtering pigs and severing the blood vessels of the throat. Immediately after the animals’ slaughter, their liver was collected. A fragment of liver tissue (from the right lobe) was immersed in a 4% buffered formalin solution for preservation. Another portion from the right lobe was placed initially in liquid nitrogen and later stored in an ultra-low-temperature freezer set at −80 °C.

### 2.3. Histology

A sample of liver tissue was removed from buffered formalin, cut into two fragments of about 1 cm^2^ in size, and then placed in histology cassettes. These cassettes were subsequently placed in a tissue processor, where they were dehydrated through increasing alcohol concentrations (30%, 50%, 70%, 80%, 96%, 100%). They were then permeabilized with xylene and embedded in paraffin. To enhance the reliability of the results, two paraffin blocks were created from each liver sample. Two slides were produced from each block by slicing the blocks with a microtome (Microm HM 340 E, Thermo Scientific, Germany) into 4 μm sections. These liver preparations were stained using the Masson-Goldner trichrome technique with aniline blue to differentiate structures. Stained slides were viewed under a light microscope (Axio Lab. A1, Carl Zeiss, Oberkochen, Germany) equipped with an Axiocam color 105 camera (Carl Zeiss, Oberkochen, Germany). The collected microscopic images underwent examination using Zen 2.3 blue edition software (version 2.3; Carl Zeiss Microscopy, Jena, Germany) and ImageJ (version 1.53; US National Institutes of Health, Bethesda, MD, USA; available at: http://rsb.info.nih.gov/ij/index.html, accessed on 15 May 2023). ImageJ software version 1.53 (US National Institutes of Health, Bethesda, MD, USA) was used to calculate the percentage of collagen fibers (colored blue: collagen fibers), with a random selection of regions of interest (ROIs) in everyone (10 ROIs). Quantitative assessment of liver fibrosis, in this case specifically peritumoral fibrosis, depended on distinguishing colors between blue (collagen fibers) and red (parenchyma). The percentage of collagen fibers in the image was assessed using the Threshold function (after setting an 8-bit color scale) [17]. The percentage of collagen fibers was counted per area of ROI (percent fiber content per area of interest). Results were reported as averages for the individual.

### 2.4. RNA Isolation and Quantitative PCR (qPCR)

RNA isolation was conducted using the Total RNA Mini (A&A Biotechnology, Gdańsk, Poland) in accordance with the producer’s recommendations. The RNA quality was assessed using the Tapestation 2200 (Agilent, Santa Clara, CA, USA), while its quantity was measured with the Nanodrop 2200 (Thermofisher Scientific, Waltham, MA, USA). For further purification, the RNAClean XP (Beckman Coulter, Beckman Coulter, Brea, CA, USA) was utilized. After purification, the RNA quality was assessed by agarose gel electrophoresis. The RNA was then reverse transcribed using the High Capacity cDNA Archive Kit (Thermofisher Scientific, Waltham, MA, USA). Subsequently, qPCR was performed using TaqMan Gene Expression Assays (Table 2): *NOTCH1 Ss03377164_u*, *CYP7A1 Ss03378689_u1*, *NR1H3 Ss03389237_g1*, *CREB1 Ss03386122_u1*, *SREBF2 Ss03376492_u1*, *SIRT3 Ss03386766_u1*, *DHCR24 Ss04323966_m1*, *APOE Ss03394681_m1*, *ACAA2 Ss04245775_m1*, *HK1 Ss04323446_m1*, *CXCL10 Ss03391845_g1*, *CYP1A1 Ss03394917_g1*, *COL1A2 Ss03375009_u1*, *SOD1 Ss03373476_u1*, and *CAT Ss04323025_m1*. The process was performed in triplicate on the QuantStudio 7-flex instrument (Thermofisher Scientific, Waltham, MA, USA) using the TaqMan Gene Expression Master Mix. For an endogenous control, *RPS29 Ss03391548_g1* was used.

Gene expression data were analyzed using the nonparametric Mann–Whitney *U* test, whereas the percentage data for collagen fibers were evaluated using one-way ANOVA. All analyses were conducted using the Statistica^®^ ver. 13.3 software package (StatSoft Inc., Tulsa, OK, USA). The experimental model comprised two groups: Group I (control) and Group II (with AST). Each piglet was considered an individual experimental unit (*n* = 8 pigs per group). Differences were deemed statistically significant when *p* < 0.05.

## 3. Results

The evidence showed that the piglets exhibited no signs of disease, indicating that the weaning procedure was normal and no undesirable health situations occurred during the rearing of the piglets. In the present experiment, we checked whether AST has any protective effect on piglets after a stressful weaning procedure.

### 3.1. Histology

Histological analysis of liver tissues showed that the addition of AST significantly reduced (*p* < 0.001) the percentage of collagen fibers around liver lobules (Figure 1). In the control group, the typical sinusoidal vessels were not present, and there was an observed alteration in the customary polygonal shape of hepatocytes, accompanied by their swelling. Conversely, in the liver tissues from the AST-treated group, the hepatocytes retained their normal shape, and the standard structure of the sinusoidal vessels was visible.

### 3.2. Gene Expression

Gene expression analysis showed that the addition of AST significantly increased the expression of *CYP7A1* (*p* = 0.001) and significantly reduced the expression levels of *CREB* (*p* = 0.040), *NOTCH1* (*p* = 0.004), and *NR1H3* (*p* = 0.002). However, there were no significant differences between the groups regarding the expression of *SREBF2*, *SIRT3*, *APOE*, *DHCR24*, *ACCA2*, *HK1*, *CXCL10*, *CYP1A1*, *COL1A2*, *SOD1*, and *CAT* (Table 3).

## 4. Discussion

Postweaning stress is a phenomenon that affects not only the behavior but also the health status of pigs. Weaning stands out as one of the most stressful periods, resulting in intestinal, immunological, and behavioral changes. During this transitional period, piglets confront numerous stressors. These include abrupt separation from the sow, transport, and handling stress, a different food source, social hierarchy stress, mixing with pigs from other litters, a different physical environment, increased exposure to pathogens and food or environmental antigens [31]. After weaning the piglets mostly return to normal status during 10–14 days, this recovery timeline can drastically extend if they encounter issues such as diarrhea, atrophy of intestinal villi, or inflammation of the intestinal mucosa. In such instances, their recuperation is protracted, feed utilization is delayed, and animal productivity declines. This underscores the idea that the repercussions of weaning may linger far longer than the effects of a singular stressor, like immobilization stress. Given the weaning procedure, the focus has shifted towards supplementary feed additives to support the health of piglets during these difficult postweaning days. The aim is to limit the negative effects of weaning on the animals’ subsequent productivity. Our task is to alleviate this transitional period as much as possible for the piglets. Our study spans 35 days and seeks to examine the effects of AST. This is examined in the context of stress induced not just by maternal separation, but related to the whole range of previously mentioned stressors.

There are limited results from detailed liver estimations. However, in the case of intensively fed and fast-growing monogastric animals, this organ is extremely loaded. The liver’s condition mirrors the health of animals subjected to poor quality or improperly balanced feed. Our research introduces a novel aspect of liver analysis, encompassing both histological and genomic analysis.

### 4.1. Histology

The study observed a potential beneficial impact of AST on the percentage of collagen in liver tissue. The analysis demonstrated that AST significantly lowered the percentage of collagen fibers in liver sections. It is vital to note that, when assessing the histological differences between the livers of the two groups, that samples were taken after 35 days into the experiment, and during the postweaning period which also affected the structure of the organ [32]. In the group fed a standard diet, irregular sinusoidal vessels were observed, along with altered hepatocyte shapes and swelling of these cells. However, in the AST-treated group, hepatocytes maintained their normal shape and the standard structure of sinusoidal vessels was evident.

Liver fibrosis is marked by the excessive buildup of extracellular matrix proteins, predominantly collagen [1]. It arises due to various diseases that inflict damage on hepatocytes, engage inflammatory cells, and activate collagen-producing cells [33]. A primary mechanism driving fibrogenesis in the liver involves the activation of HSCs following organ damage. Upon activation, HSC transforms from a resting state to a proliferative, contracting myofibroblast-like cells, producing type I collagen—one of the extracellular matrix’s primary components [34,35,36]. This stellate cell activation can be exacerbated by oxidative stress [37]. Oxidative radicals, through interference with mitochondrial function, can catalyze hepatocyte death (oncotic necrosis, necroptosis and apoptosis) and promote the secretion of proinflammatory mediators [10]. Hence, it is proposed that Reactive Oxygen Species (ROS) could encourage fibrosis by harming hepatocytes or by activating Kupffer cells and HSCs [11,34]. Recent studies underscore AST’s positive effects on liver health. It is mainly its antioxidant properties that allow AST to play a protective role [1]. Other scientific reports indicate a controlling effect of AST against glycemia in prediabetes patients at risk for metabolic syndrome with mixed effects on lipid levels [12,13,35,38]. Yang et al. (2016) [14] showed that astaxanthin inhibits the activation of resting-phase HSCs and restores the resting state of activated HSCs in mice. AST also decreased ROS production and elevated the expression of the nuclear erythroid-related factor 2 (NrF2).

### 4.2. Gene Expression

The study also investigated AST’s impact on the expression of specific genes, emphasizing the liver’s pivotal role in lipid metabolism. Lipid metabolism within hepatocytes can be divided into three stages: I—lipid acquisition and fatty acid synthesis (de novo lipogenesis); II—lipid storage; III—lipid consumption, including lipolysis, β-oxidation, and secretion of low-density lipoproteins [36,39]. Many studies have shown that genes involved in metabolism are overexpressed or under expressed based on dietary factors.

Our analysis showed that the expression level of *CYP7A1* increased significantly in the AST-treated group. Cholesterol 7α-hydroxylase (*CYP7A1*) is the rate-limiting enzyme in the classical bile acid synthesis pathway. These findings align with those of Liu et al. (2022) [40], who found that AST enhances the fecal excretion of both acidic and neutral sterols derived from cholesterol by increasing *CYP7A1* expression. Similarly, a study by Wang et al. (2022) demonstrated that administering a high dose of AST (0.75%) to mice on a high-fat diet elevated *CYP7A1* expression, compared to mice only on a high-fat diet [41]. From a physiological perspective, the upregulation of *CYP7A1* is advantageous, facilitating cholesterol efflux which protects against excessive increases in plasma cholesterol concentrations.

Our study observed notably diminished levels of *NR1H3* expression in the AST-treated group. *NR1H3* is part of the LXR nuclear receptor superfamily [42], which oversees cholesterol equilibrium, lipoprotein metabolism and fat synthesis [42,43,44]. Notably, *NR1H3* signaling is associated with liver diseases such as hepatic steatosis. In addition, it contributes to liver triglyceride content by upregulating lipogenic genes [40,45]. Zhang et al. [45] found a correlation between *NR1H3-exon-5-A201C* and fat thickness, emphasizing that *NR1H3* gene expression can modulate lipid accumulation in pigs. Studies have shown that LXRs are prevalent nuclear receptors in HSC. Signaling via these receptors adjusts the gene expression related to metabolism, inflammation and fibrogenesis in primary cells [46,47]. Beaven et al. [47] exemplified this with their discovery that LXR ligands inhibited fibrosis markers and stellate cell activation in primary mouse stellate cells. In the present study, there is a noticeable association between the reduced expression of *CREB* and the decrease in liver tissue collagen percentage in the AST-supplemented group. *CREB* influences stellate cell activation and proliferation, as well as inflammation. Furthermore, studies indicate that *CREB* is involved in extracellular matrix deposition and functional cell loss [48]. Eng and Friedman (2001) [49] posited that, under oxidative stress, a mutated form of *CREB* (CREB Ala 133) led to a surge in stellate cells transitioning to the S phase, initiating HSC proliferation sequences. Moreover, *CREB*’s protective role against hepatocyte apoptosis linked to mitochondria, mediated via the *ERK-CREB-Bnip3* axis in hepatic steatosis, has been documented [50]. Wang et al. (2016) emphasized *CREB*’s association with liver fibrosis, revealing enhanced p-*CREB-1* expression in fibrotic liver tissues in rats and HSCs exposed to exogenous *TGF-β1*, suggesting *p-CREB*-1 might boost fibrogenesis by enhancing *TGF-β1* expression in liver fibrosis [51]. Furthermore, we noted a marked decrease in *NOTCH1* expression in the AST-supplemented pigs. *NOTCH* signaling pathway plays an important role in adipogenesis processes, such as the proliferation and differentiation of adipocyte progenitors, and is integral to metabolism regulation [52,53]. Yamaguchi et al. [54] showed that *NOTCH* haploinsufficiency heightens fat accumulation and adipogenesis, underscoring the tie between *NOTCH* signaling and insulin resistance onset. In mice fed a high-fat diet, *NOTCH* signaling augments fat storage in the hypothalamus via B cell (NF-κB) activation [55]. *NOTCH* signaling’s activation in liver fibrosis models and its abnormal amplification in fibrosis patients have been documented, while inhibition of *NOTCH* has shown protective effects against fibrotic disorders [56]. In our experiment, we also analyzed several liver function-related genes (*APOE*, *ACAA2*, *CYP1A1*, *SIRT3*, *HK1*, *DHCR24*, and *CXCL10*). However, after AST addition, no significant expression changes were identified. These genes encompass a variety of functions: lipid metabolism (*APOE* [57], *ACAA2* [58,59]), exogenous chemical metabolism (*CYP1A1* [60]), mitochondrial fatty acid oxidation (*SIRT3* [61]), antioxidation, free radical scavenging, glucose metabolism (*HK1* [62]), cholesterol synthesis regulation (*SREBF2*, *DHCR24* [63,64]), cellular stress responses (*DHCR24* [63,64]), and liver inflammation augmentation (*CXCL10* [65]). In the case of *CXCL10*, the important information from the point of view of this work is that it promotes liver fibrosis by preventing NK cell-mediated inactivation of hepatic stellate cells [66,67].

Fibrosis markers, such as *COL1A2* [65,66], encode collagen. Both *COL1A1* and *COL1A2* genes exhibit a high sensitivity to reactive oxygen species [68]. It is postulated that under oxidative stress, *COL1A2* mRNA expression escalates, a factor contributing to stellate cell activation [67]. Thus, dampening *COL1A2* expression might inhibit stellate cell activation, offering a potential preventive measure against liver fibrosis [68]. In the present study, we observed no significant changes in *COL1A2* expression in both groups. Similarly intriguing is the absence of significant changes in the expression of *SOD1* and *CAT*. While some literature posits that AST may boost hepatic antioxidants *SOD* and *CAT* through partial induction of the Nrf2 pathway [63]. The increase in *SOD* after AST treatment has been confirmed in a few studies on hens [69], rats [70] and aquatic animals [71]. However, the methods for studying the activity or concentration of *SOD* have varied. Islam et al. [72] proved that the addition of AST could restore *CAT* and *SOD* activity in rats with carbon tetrachloride-induced liver fibrosis. They argued that AST’s potential protective mechanism stems from lipid peroxidation inhibition and stimulating cells of antioxidant systems. We speculate that the lack of change in *SOD* and *CAT* expression in the experiment may be due to insufficient administration of (AST) to the experimental animals.

The obtained findings provide compelling evidence supporting the potential antifibrotic properties of AST, particularly when considering the age of the experimental animals and their weaning period. Nevertheless, the authors acknowledge the constraints of this study, and they propose the need for additional analyses to further explore the effects of astaxanthin on the liver. These suggested investigations encompass immunohistochemical analyses of HSC activity, assessments of serum redox states, and detailed quantifications of interleukin and proinflammatory cytokines. The authors already have planned future studies to expand upon the evidence presented in this study, ensuring a more comprehensive understanding of astaxanthin’s impact on liver health.

## 5. Conclusions

In the experiment, significant differences in histology between animals with and without astaxanthin in the diet were observed. In the group supplemented with AST, there was a noticeable reduction in collagen fiber deposition in liver sections, suggesting AST’s potential role in safeguarding against excessive collagen accumulation. Furthermore, piglets on the astaxanthin regimen exhibited a decline in *NR1H3* expression—a gene whose inhibition can foster recovery from hepatocyte steatosis, and increased expression of *CYP7A1*—a pivotal gene for expelling surplus liver cholesterol through bile acids. Additionally, the downregulation of *NOTCH1* and *CREB* genes was observed, although their exact roles in mitigating liver fibrosis are ambiguous. Gene expression analysis has shown the potential of AST to protect the liver, but further studies are needed to understand the molecular basis of this action.

## Figures and Tables

**Figure 1 animals-13-03268-f001:**
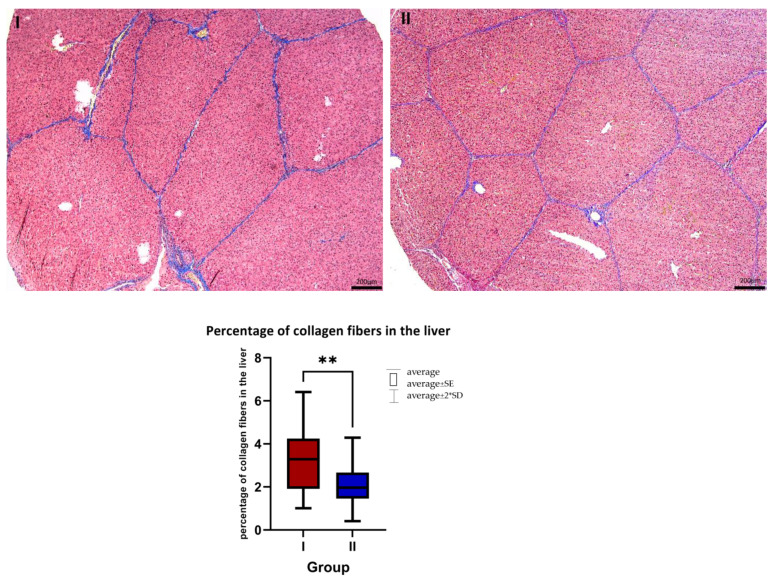
Masson-Goldner staining with aniline blue. I—control group, II—a group with AST addition. Magnification 5×. Scale bar = 200 μm. The average percentage of collagen fibers in the liver in the control group (I, *n* = 8) and the AST-treated group (II, *n* = 8). Mean ± SD (I) = 3.412 ± 1.65%; mean ± SD (II) = 2.018 ± 1.05%. ** *p* ≤ 0.01.

**Table 1 animals-13-03268-t001:** Ingredients (%) and nutritive value of diets in the experiment.

	I	II
Items	Control	AST
Soybean pressed cake	19	19
Wheat	42.17	42.17
Corn	20	20
Rapeseed oil	0.8	0.8
Skimmed milk powder	10	10
Dried whey	5	5
1-Ca phosphate	0.4	0.4
Feed chalk	1.15	1.15
Salt	0.13	0.13
Lysine	0.4	0.4
Methionine	0.23	0.23
Threonine	0.16	0.16
Tryptophan	0.06	0.06
Vitamin-mineral premix *	0.5	0.5
Astaxanthin (0.025 g/kg)	-	+
Content in 1 kg:		
Dry matter, g	901	901
Crude protein, g	183	183
Crude fat, g	40	40
Crude fiber, g	27	27
Crude ash, g	56	56
Metabolizable energy, MJ **	13.8	13.8
Lysine, g	13.7	13.7
Methionine + Cystine, g	8.2	8.2
Threonine, g	8.8	8.8
Tryptophan, g	2.7	2.7
Calcium, g	7.6	7.6
Phosphorus digestible, g	3.1	3.1

AST—astaxanthin; * Content in 1 kg of premix: vit A—2,400,000 IU; vit D3—400,000 IU; vit E—8000 IU; vit B1—400 mg; vit B12—6000 µg; vit B2—1000 mg; vit B5—3000 mg; vit B6—600 mg; vit K—400 mg; biotin—30,000 µg; niacin—5008.3 mg; folic acid—100 mg; pantothenic acid—2760 mg; choline—24,193.548 mg; betaine—12,000 mg; Cu—20,000 mg; Fe—20,000 mg; I—200 mg; Mn—8000 mg; Se—60 mg; Zn—24,000 mg; Ca—267.979 g; Cl—6.268 g; K—0.066 g; Mg—30 g; Na—0.037 g; S—22.245 g. ** Metabolizable energy calculated using the equation of Hoffmann and Schiemann [30].

**Table 2 animals-13-03268-t002:** TaqMan assays that were used in the study.

Gene		Primers Sequence	Amplicon Length, bp
*RPS29*	ribosomal protein S29	Ss03391548_g1	71
*HK1*	hexokinase 1	Ss04323446_m1	59
*NOTCH1*	notch receptor 1	Ss03377164_u	71
*CYP7A1*	Cytochrome P450 family 7 subfamily A member 1	Ss03378689_u1	119
*NR1H3*	Nuclear receptor subfamily 1 group H member 3	Ss03389237_g1	101
*CREB1*	cAMP responsive element binding protein 1	Ss03386122_u1	98
*SREBF2*	Sterol regulatory element binding transcription factor 2	Ss03376492_u1	67
*SIRT3*	Sirtuin 3	Ss03386766_u1	80
*DHCR24*	24-dehydrocholesterol reductase	Ss04323966_m1	104
*APOE*	Apolipoprotein E	Ss03394681_m1	63
*ACAA2*	Acetyl-CoA acyltransferase 2	Ss04245775_m1	71
*CXCL10*	C–X–C motif chemokine ligand 10	Ss03391845_g1	146
*CYP1A1*	Cytochrome P450 family 1 subfamily A member 1	Ss03394917_g1	77
*COL1A2*	Collagen type I alpha 2 chain	Ss03375009_u1	76
*SOD1*	Superoxide dismutase 1	Ss03373476_u1	77
*CAT*	Catalase	Ss04323025_m1	56

**Table 3 animals-13-03268-t003:** Analysis of gene expression in group I and group II. RQ, relative quantity of mRNA in comparison to housekeeping gene.

Gene	Mean RQ ± SE		*p*-Value
	Group		
	I	II	
*CYP7A1*	0.403 ± 0.07	0.927 ± 0.14	0.001
*SREBP2*	0.881 ± 0.11	0.786 ± 0.14	0.232
*SIRT3*	0.210 ± 0.05	0.395 ± 0.11	0.279
*APOE*	0.439 ± 0.11	0.564 ± 0.22	0.867
*CREB*	1.701 ± 0.32	0.807 ± 0.07	0.040
*NOTCH1*	0.433 ± 0.07	0.193 ± 0.03	0.004
*NR1H3*	0.459 ± 0.08	0.134 ± 0.05	0.002
*DHCR24*	0.268 ± 0.04	0.185 ± 0.04	0.121
*ACCA2*	0.196 ± 0.24	0.127 ± 0.13	0.878
*HK1*	1.271 ± 0.24	1.183 ± 0.14	0.959
*CXCL10*	0.205 ± 0.04	0.158 ± 0.04	0.152
*CYP1A1*	3.134 ± 1.28	6.112 ± 4.11	0.645
*COL2A1*	3.147 ± 0.50	3.053 ± 0.52	0.954
*SOD1*	1.326 ± 0.18	1.638 ± 0.23	0.382
*CAT*	2.381 ± 0.51	2.323 ± 0.34	0.878

## Data Availability

The data supporting reported results are in the possession of the authors (K.S., M.O. and M.Ś.).

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
