# Peer review of "The Protective Effects of Astaxanthin (AST) in the Liver of Weaned Piglets"

_animals, 2023, doi:10.3390/ani13203268_

Round 1

Reviewer 1 Report

Review:

The paper is very interesting and timely. In intensive pig production, great emphasis is placed on the number of piglets weaned per litter and on the prolificacy of sows. During the weaning period, piglets under stress often have problems with the digestive system. Their nutrient absorption functions are impaired, which translates into poorer results in the growth and development of young pigs.

The additive Astaxantin (AST) used in the study in feed for weaned piglets prevents liver fibrosis by reducing collagen deposition. According to the authors, further research must be continued to better understand the molecular basis of AST.

Comments:

 L 32 is "PBZ" should be "PL" (Polish Landrace)

Chapter: Materials and Methods – there was no information about microclimatic conditions in the building in which the experiments were carried out. What was the temperature and moisture and airflow?

Author Response

We are very grateful for all valuable suggestions. We have tried to consider all comments and their explanations have been used to improve our manuscript.

1. L 32 PBZ should be;  PL; (Polish Landrace) - this is done (line 32)
2. Chapter: Materials and Methods – there was no information about microclimatic conditions in the building in which the experiments were carried out. What was the temperature and moisture and airflow? - this is done (lines 144-149)

Reviewer 2 Report

In this manuscript, the authors investigated the protective properties of AST in the liver of weaned piglets. They found that treatment of AST resulted in the reduction of the percentage of collagen in liver tissue and AST addition affected the expression level for a few genes. Overall, the manuscript’s conclusion is useful for the pig industry, a few concerns need to be addressed before it can be accepted.

1. Line 121-122, the authors should explain why they used 0.025g astaxanthin per 1kg of feed mixture, without an explanation in somewhere, this is too arbitrary.

2. Merge Photo1 and Graph1 together, and name it as Figure1.

3. The authors should show all of the gene expression data within one figure, rather than use both table S1 and Graph2 and the data showed in Graph2 is same with table S1, but with different presentation methods, this should avoid.

4. Change the name of Graph2 to Figure2.

Author Response

We are very grateful for all valuable suggestions. We have tried to consider all comments and their explanations have been used to improve our manuscript.

1. Line 121-122, the authors should explain why they used 0.025g astaxanthin per 1kg of feed mixture, without an explanation in somewhere, this is too arbitrary. – this is done (lines 124-128)

2. Merge Photo1 and Graph1 together, and name it as Figure1 - this is done.

3. The authors should show all of the gene expression data within one figure, rather than use both table S1 and Graph2 and the data showed in Graph2 is same with table S1, but with different presentation methods, this should avoid – this is done.

4. Change the name of Graph2 to Figure2 - We decided to remove Graph 2 and present Table 3.

Reviewer 3 Report

The manuscript investigates the protective properties of AST in the liver of weaned piglets based on expression of genes and histological assessment of liver tissues. L117 states 35-day old post-weaned pigs. How long after weaning was this? (14 days, 7 days)? This could have considerable consequence for the intention to assess the effect of AST during weaning-related oxidative stress. Tissue was collected at 35 days after start of diets. There is no indication that the pigs underwent any kind of particular stress (i.e. rearrangement of pen mates, weaning, disease challenge, environmental temperature) during the period of feeding experimental diets. If pigs were weaned at 21 days of age, it is likely there was considerable recovery from that stressor prior to trial start and no real indication of additional ‘stress’ during the trial feeding period. No data is presented to demonstrate a difference in oxidative status, stress, health, etc between the groups. It is presumed that this work was part of a larger trial reported in the reference to “earlier data” in L 96 (ref. 24). In this reference, the authors state pigs were “healthy during the experiment with no sign of disease” which matches with the lack of difference in growth performance reported and supports the questions above relating to likelihood to associate reported differences to AST’s protective effect (protective of what exactly). The ‘control’ pigs do not appear to have been under any kind of challenge that required protection from something. There is minimal data to support the hypothesis and if this was part of the previously reported study, could have been included in that publication rather than as a stand alone study.

Other specific comments

-       Graph 2 represent relative fold change in comparison to the housekeeping gene or something else? or what does RQ represent? 

Minor revisions required

Author Response

We are very grateful for the evaluation. We will try our best to answer your concerns.

Reviewer 3: The manuscript investigates the protective properties of AST in the liver of weaned piglets based on expression of genes and histological assessment of liver tissues.
L117 states 35-day-old post-weaned pigs. How long after weaning was this? (14 days, 7 days)? This could have considerable consequences for the intention to assess the effect of AST during weaning-related oxidative stress. Tissue was collected 35 days after the start of the diet. There is no indication that the pigs underwent any kind of particular stress (i.e.rearrangement of pen mates, weaning, disease challenge, environmental temperature) during the period of feeding experimental diets. If pigs were weaned at 21 days of age, it is likely
there was considerable recovery from that stressor before the trial started and no real indication of additional ‘stress’ during the trial feeding period. No data is presented to demonstrate a difference in oxidative status, stress, health, etc between the groups. It is presumed that this work was part of a larger trial reported in the reference to “earlier data” in L 96 (ref. 24). In this reference, the authors state pigs were “healthy during the experiment with no sign of disease” which matches with the lack of difference in growth performance reported and supports the questions above relating to likelihood to associate reported differences to AST’s protective effect (protective of what exactly). The ‘control’ pigs do not appear to have been under any kind of challenge that required protection from something. There is minimal data to support the hypothesis and if this was part of the previously reported study, could have been included in that publication rather than as a stand-alone study.
Answer: The pigs were weaned at 35 days of age (not at 21 days of age), so the experiment began on the day of weaning and continued for another 35 days. The stressor, in this case, was “weaning” - the standard farm procedure, and thus: separation from the sow, change of location and environment, placement in pens, and change of feeding. It is known that this procedure is a multi-type stress, and its effects are reflected in the health and productivity of piglets for several consecutive days. All of the piglets – control and experimental ones – are
under the same multi-type stress (weaning). If this procedure cannot be omitted, feed
additives are used to support the health of piglets through these difficult days after weaning,
and to limit the negative effects of weaning on the subsequent productivity of the animals. Hence, we consider it reasonable to check the protective effect of AST in these pigs. In our experiment all piglets passed the stressful procedure of weaning, however, the experimental animals obtained the AST supplementation, while the control group was devoid of any feed additive. This study was aimed at testing whether astaxanthin, which is one of the most potent antioxidants, would benefit the liver in weaned piglets. We agree that after 35 days, pigs could recovery from the stress caused by weaning. However asataxantine was provided to the experimental group immedietly after the weaning, thus our aim was to observe if the asataxantine helps in this recovery.
Yes, the piglets in this experiment were part of a large project partly described in the publication: Szczepanik, K.; Furgał-Dierżuk, I.; Gala, Ł.; Świątkiewicz, M. Effects ofHermetia Illucens Larvae Meal and Astaxanthin as Feed Additives on Health and Production Indices in Weaned Pigs. Animals 2023, 13, doi:10.3390/ani13010163. Due to the complexity and multifactoriality of the whole project in reference (24), it is not possible to combine these publications into one article. We have evidence that the piglets showed no signs of disease, which means that the weaning procedure was normal and no undesirable health situations occurred during the rearing of the piglets. In our experiment, we checked whether AST has any protective effect on piglets after stressful weaning (but healthy piglets, because the AST is a dietary supplement, not a medicine).

Reviewer 3: Graph 2 represents relative fold change in comparison to the housekeeping gene or something else? or what does RQ represent? 

Answer: The “RQ” represents the Relative Quantification. The relative quantity (RQ) of each sample was calculated based on the ΔΔCt method using QuantStudioTM 6 and 7 flex real-time PCR software. The RQ is a relative fold change in comparison to housekeeping gene –corrected in lines 228-229.

Round 2

Reviewer 3 Report

Clarifications within the paper are helpful. The statement the study demonstrates 'protective' effect is still not well justified. The authors state the relevant stress in this study was weaning and that this is relevant to the test additive because it is "supplement" not "medicinal". Good point, the problem is the time of collection does not line up with the known period of stress after weaning (i.e. within the first 10 days, maybe out to 14 days). The authors note "its [weaning] effects are reflected in the health and productivity of the piglets for several consecutive days". 35 days after weaning is no longer "several consecutive days". There is no evidence in the introduction or elsewhere that demonstrates effects of weaning on the liver for this length of time. Certainly, data in the gut has demonstrated impacts of weaning are no longer evident within  4 week of weaning under 'normal' weaning conditions which is what occurred in this case. This work can demonstrate a low risk of detrimental effect of feeding AST to weaning pigs but the data does not support 'protective' effect.

Author Response

Revewer 3

Thank you very much for this valuable comments which allow us to better discuss our results and improve our manuscript. We will try to dispel any doubts and clarify some information.

  1. Reviewer: The authors state the relevant stress in this study was weaning and that this is relevant to the test additive because it is "supplement" not "medicinal". Good point, the problem is the time of collection does not line up with the known period of stress after weaning (i.e. within the first 10 days, maybe out to 14 days). The authors note "its [weaning] effects are reflected in the health and productivity of the piglets for several consecutive days". 35 days after weaning is no longer "several consecutive days".

Answer: This is true that 10-14 days after weaning the piglets mostly return to normal, provided that in the meantime they did not suffer from diarrhea and there was no atrophy of the intestinal villi or inflammation of the intestinal mucosa, because then the recovery takes longer, good feed utilization is delayed, and animal productivity declines. Our task is to alleviate this transitional period as much as possible. The research on the effect of feed mixtures must meet certain conditions, and one of them is to maintain an appropriate introductory period during which the piglets get used to the new feed mixture. Normally for monogastric animals it is from 3 to 10 days [Kong and Adeola, 2014*; Zhang and Adeola, 2017**], usually 7 days is practiced. From our experience the time at which this experiment was conducted (35 days) is optimal when using experimental supplements and feed additives. Post-weaning stress is a phenomenon that affects not only the behavior but also the health status of pigs. Weaning is one of the most stressful periods, resulting in intestinal, immunological and behavioral changes. During this period, young animals are exposed to a number of stressors, such as sudden separation from the sow, transport and handling stress, a different food source, social hierarchy stress, mixing with pigs from other litters, a different physical environment, increased exposure to pathogens and food or environmental antigens [Campbell et al., 2013***]. Thus, it can be thought that its effects and impact are evident over a much longer period of time than if one particular stressor (e.g., immobilization stress) is acted upon. The 35-day experimental period is intended to examine the effects of AST on the piglet's body induced by stress understood not only through separation from the mother, but related to the whole range of previously mentioned stressors. In Poland, in practice, piglets are weaned at 4 or even 5 weeks of age, depending on the farm. However, according to many sources, more important than the weaning age is the animal's weight, which should be 7-8 kg. To confirm the length of the experiment and the weaning day, we present the following publications:

  • Jin, X. H., Heo, P. S., Hong, J. S., Kim, N. J., & Kim, Y. Y. (2016). Supplementation of dried mealworm (Tenebrio molitor larva) on growth performance, nutrient digestibility and blood profiles in weaning pigs. Asian-Australasian Journal of Animal Sciences, 29(7), 979.
  • Ferreira, A. S., Barbosa, F. F., Tokach, M. D., & Santos, M. (2009). Spray dried plasma for pigs weaned at different ages. Recent Patents on Food, Nutrition & Agriculture, 1(3), 231-235.
  • Li, D., Liu, S. D., Qiao, S. Y., Yi, G. F., Liang, C., & Thacker, P. (1999). Effect of feeding organic acid with or without enzyme on intestinal microflora, intestinal enzyme activity and performance of weaned pigs. Asian-Australasian Journal of Animal Sciences, 12(3), 411-416.

  1. Reviewer: There is no evidence in the introduction or elsewhere that demonstrates effects of weaning on the liver for this length of time.

Answer: Unfortunately, there are not many results from detailed liver estimations, and yet in the case of intensively fed and fast-growing monogastric animals, this organ is extremely loaded. We think it's worth studying. The liver reflects the health of animals when using poor quality or poorly balanced feed. In our research the liver analysis have an novelty aspect, especially as it includes both histological and genomic analysis.

  1. Reviewer: This work can demonstrate a low risk of detrimental effect of feeding AST to weaning pigs but the data does not support 'protective' effect.

Answer: We used the Reviewer's point of view to refine and improve our Conclusions (corrected lines 354-364). In the presented experiment the addition of AST statistically significantly reduced the percentage of collagen fibers around the liver lobules. In addition, the structure of the organ was also improved, which was evident in the corrected shape of hepatocytes and the regular structure of sinusoidal vessels. Piglets receiving AST showed reduced expression of NR1H3, a gene whose silencing promotes recovery from hepatocyte steatosis, and increased expression of CYP7A1, a gene responsible for removing excess cholesterol from the liver via bile acids. All these above mentioned results have been confirmed statistically. The results showed that beneficial changes in liver structure were observed in piglets receiving astaxanthin supplementation. In our opinion this proves that AST can show the protective effect in weaned piglets feeding.

*Kong and Adeola (2014). Evaluation of amino acid and energy utilization in feedstuff for swine and poultry diets. Asian Australas. J. Anim. Sci. 27:917-925.

**Zhang F. and Adeola O. (2017). Techniques for evaluating digestibility of energy, amino acids, phosphorus, and calcium in feed ingredients for pigs. Animal Nutrition 3, 344-352.

***Campbell, J. M., Crenshaw, J. D., & Polo, J. (2013). The biological stress of early weaned piglets. Journal of animal science and biotechnology, 4(1), 19.